# Lab-on-a-Chip Technologies for the Single Cell Level: Separation, Analysis, and Diagnostics

**DOI:** 10.3390/mi11050468

**Published:** 2020-04-29

**Authors:** Axel Hochstetter

**Affiliations:** Experimentalphysik, Universität des Saarlandes, D-66123 Saarbrücken, Germany; axel_hochstetter@web.de; Tel.: +49-(0)681-302-2730

**Keywords:** microfluidics, single cell level, diagnostics, biomedical engineering, parasites, cancer, infectious diseases, point-of-care

## Abstract

In the last three decades, microfluidics and its applications have been on an exponential rise, including approaches to isolate rare cells and diagnose diseases on the single-cell level. The techniques mentioned herein have already had significant impacts in our lives, from in-the-field diagnosis of disease and parasitic infections, through home fertility tests, to uncovering the interactions between SARS-CoV-2 and their host cells. This review gives an overview of the field in general and the most notable developments of the last five years, in three parts: 1. What can we detect? 2. Which detection technologies are used in which setting? 3. How do these techniques work? Finally, this review discusses potentials, shortfalls, and an outlook on future developments, especially in respect to the funding landscape and the field-application of these chips.

## 1. Introduction

Since the advent of microfluidics 30 years ago, many applications have employed the advantages of microfluidic environments: small sample volumes, ready parallelization, high reproducibility, a vast span of experimental timescales, and a high control over local conditions (e.g., temperature, light exposure, flow velocity and direction, shear forces, diffusion, concentration gradients, viscosity, cell motility, etc.). One very prominent aspect of microfluidic research is diagnostics on the single-cell or even molecular level. The significance of recent research towards microfluidics-based single cell diagnostic chips is apparent in health care, in our homes, and also very prominently in the fight against the COVID 19 pandemic: The diagnostic targets range from circulating tumor cells (CTC) [1,2,3,4,5,6], over parasites in blood [7,8,9,10,11,12,13,14,15], male fertility [16,17,18,19,20], molecular markers for infections [11,15,21,22,23], cells of a specific stage in their life cycle [24,25], plant pathogens [26] and the SARS-CoV-2 proteome [27,28,29,30,31,32]. Depending on the exact target (either the entire cell or sub-cellular markers) there are different approaches to on-chip detection, each with their own underlying fundamentals and set of limitations. Additionally, there are additional synergetic possibilities (smartphones, optical traps, high throughput, personalized test, portability). This review presents a collection of the techniques proven useful for single cell diagnostic chips and explains the basic physical, chemical and biological effects that drive these technologies.

Due to the small volumes and the readily available lithographic procedures, diagnostic chips often are mass-producible, which makes individual tests cheap. This also opens up a great potential for portable diagnostics for global health issues [33] even in rural and remote locations, like endemic areas in Africa. Sadly, and very surprisingly, this potential often is not reached, and many sound diagnostic devices end in the “valley of death” [34,35]. In my opinion, this is due to a hole in the funding landscape; while many funding agencies (e.g., the Bill & Melinda Gates Foundation, UKRI, HFSP) offer grants to research the technologies and their application in a lab, the time required to actually adapt these technologies into field-applicable devices extends beyond the general timeframe of these grants and there are by far not enough grants that cover the adaption and deployment of field-ready devices. Furthermore, the adaptation of devices to make them field-applicable requires an inter- and cross-disciplinary skillset and contacts to both relevant populations and health-system officials. These requirements for device adaptation and deployment are massively at odds with how a research group has to be structured to be successful in today’s academic and funding landscape. Sadly, Academia in most countries selects for high-throughput high-impact publishing researchers, while these Grand Challenges tend to be tackled by researchers who follow a conviction to better the world, and sacrifice their publication output along the way. If mankind really aims to not just face, but succeed in the face of these Grand Challenges, Academia needs to adapt, raise their standards beyond the number of the h-index.

### What We Can Detect:

In recent years, more and more targets for detection on diagnostic chips have been investigated. Some salient examples thereof are circulating tumor cells (CTC) of various types of cancer [1,36,37,38], rare cells (e.g., sickle-cell variants of red blood cells) [39,40], parasites, like *Plasmodium falciparum* [1,7,10,11,13,14,15,21,22,23,36,41,42,43,44,45,46,47,48,49,50] and *Trypanosoma spp.* [8,44,51,52,53,54,55,56,57] and even plant pathogens [26,58], as well as—after cells have been lysed—subcellular infection markers (e.g., DNA, RNA fragments) [10,11,22,43,59,60,61,62]. Given the vast adaptability of microfluidics to any kind of single or multi-cellular assay [63], the ability to combine it with various light microscopy techniques [64], image processing [65], optical or acoustic traps [53], generation of chemical gradients [66], and even cell culture [4,67,68,69,70,71,72,73,74,75,76,77,78,79,80,81,82,83], any cellular or subcellular target seems to be possible for future on-chip diagnostics.

For easier access to the contents of this review, please find below a table which summarizes all the techniques discussed throughout this publication and their applications toward single cell diagnostic chips and beyond (see Table 1).

## 2. Methodologies

For this review, all results in Google Scholar for literature related to “single cell diagnostic chips” and their permutations of the last five years were screened. Additionally, where technologies were mentioned, the relevant original research papers pertaining to these technologies were screened, cited, and summarized to yield a more rounded review. The findings are summarized below in three chapters:Cell separation methodsCombined separation and analysis on chipMolecular analysis of single cells

### 2.1. Cell Separation Methods

In the last three decades, since the advent of microfluidics, several methodologies have been developed to create single cell diagnostic chips. While these three paragraphs only list these methodologies, the next section of this publication describes each of these methodologies and the underlying principles or technologies. For separation of cells in a microfluidic setup, we can either work by encapsulating samples inside droplets, or by leaving the entire sample inside a continuous (aqueous) phase. If the sample is in the continuous phase, we can separate the target cells either using deterministic lateral displacement (DLD), ratchets, dean-flow, di-electrophoresis, surface acoustic waves (SAW), optical and acoustic tweezers or by using optical density/refractive index. In a continuous phase it is also possible to follow a temporal evolution along the flow of the device.

By introducing two immiscible phases (e.g., water and oil) it is possible to create droplets, inside which the sample can be portioned. Droplet microfluidics has so far been used together with di-electrophoresis, deterministic lateral displacement, surface acoustic waves, or hydrodynamic droplet sorting to separate cells. It also has great potential, because it can be used to generate vast libraries of individual reaction volumes. This makes it possible to screen how identical cells react to entire sets of parameters in one single setup [84,85]. This can also be used to parallelize the analysis or separation of several sets of samples.

In case the target is not the entire cell, but a sub-cellular marker (e.g., DNA or RNA, individual antibodies [62] or other molecules) the cells containing the target molecule are lysed and/or the non-relevant cells are separated. After lysing the cells, their contents are optionally selectively multiplied (using e.g., polymerase chain reaction, PCR, and related techniques, like qPCR and nested PCR or Loop-mediated Isothermal Amplification, LAMP, and its derivatives like NINA-LAMP and LAMPport) and analyzed using chromatographic approaches (e.g., metabolomics, transcriptomics, proteomics, polyomics) or antibody-antigen binding, especially in rapid diagnostic tests (RDTs) [62]. This can all be done either in continuous phase microfluidics, droplet microfluidics, or paper-based microfluidics.

#### 2.1.1. Dielectrophoresis (DEP)

Dielectrophoresis (DEP) describes the movements of cells (and other dielectric particles) within in a non-uniform electric field. This movement is caused by the induction of an electric dipole moment on the cell or particle, and the force the electric field gradient exerts on this dipole moment. It is not necessary that a cell carries a net surface electric charge to be polarized [8]. The induced dipole is aligned along the gradient of the non-uniform electric field. Thus, the coulomb forces generated on both sides of the dipole/particle are different, and a net force drags the particle across the gradient. The direction and strength of this force depends on many factors, both of the particle itself and the surrounding medium including the particles exact shape [86] and the frequency of the alternating current (AC) electric field [54]. For more details, see a recent review by Adekanmbi and Srivastava [86].

In short: by varying the frequencies of the employed AC electric field and/or the conductivity of the surrounding medium (e.g., by adding salts to increase the medium’s conductivity), different cells can be separated. A prominent example is the separation of red blood cells (RBCs) and trypanosomes (a unicellular pathogenic parasite), conducted by Menachery *et al.* [54]. They used a micro-fabricated gold four-arm spiral quadrupole electrode array (see Figure 1 and Figure 2), with each arm arranged at 90° to each other and separated by 400 µm operated at frequencies ranging from 10 kHz to 400 kHz and with solution conductivities varying from 16 to 60 mS/m. 

Within this setup, it was possible to separate trypanosomes from murine RBCs at 140 kHz, and from human RBCs at 100 kHz and a Voltage of 2 V peak-to-peak, respectively [54]. This demonstrates that it is possible to completely separate different cell types from the same sample, simply based on their induced dipole moment. Since the induced dipole moment is specific for healthy cells (e.g., RBCs), infected cells (e.g., RBCs infected by *Plasmodium falciparum*), and pathogenic cells (e.g., trypanosomes) they can be separated in such an experimental setup [6,8,54,86,87] (see Figure 2). In many cases, the detection of a single infected cell or individual pathogen within a real-life sample can arguably be counted as a diagnosis of the respective infection (here: Malaria and Trypanosomiasis, respectively). 

DEP can also be combined with other technologies for single cell diagnostic chips. One example is the combination of DEP with deterministic lateral displacement (DLD) as demonstrated by Jason Beech and others [6,8,88,89,90]; another prime example is the combination with SAW (see Section 2.1.5.), as used by Smith *et al.* to extract viable mesenchymal stroma cells from human dental pulp [91]. A drastically different usage of DEP is shown by Noghabi *et al.*, who developed a same-single-cell analytic DEP chip, to study multidrug resistance inhibition in leukemic cells [92]. To achieve higher throughput, Faraghat *et al.* used three-dimensional DEP electrodes featuring tunnels, along which the cells were separated in a more continuous fashion [93].

For an overview of the pros and cons of DEP in the context of single cell separation and diagnostics, refer to Table 2.

#### 2.1.2. Deterministic Lateral Displacement (DLD)

One core microfluidic technique used to separate cells by their size and shape (or rather, their effective hydrodynamic diameter) is deterministic lateral displacement (DLD), which was discovered accidently by Huang in 2004 [94]. There, laminar flow through a repetitive array of obstacles resulted in an asymmetrically bifurcated flow. This asymmetric flow separated different particles according to their diameter. In the 15 years since this empirical description of a phenomenon, additional aspects of DLD have been developed and have broadened its applications: the continuous separation of particles, cancer [3,95,96], healthy blood cells [40,97], infected RBCs [98], yeast [99], bacteria [24], fungal spores [100] and subcellular particles like DNA [101], and virus capsids [102]. Additionally, antibody-coated DLD arrays have been used for non-invasive prenatal diagnosis of circulating fetal cells in samples of their mother’s blood [60]. Similarly, Hou *et al.* reported an antibody-coated “nanoVelcro” assay that selectively retains circulating fetal nucleated cells from blood samples of pregnant women [103].

The basic model that is used to describe how DLD works, is referred to by experts as a “naïve model”, as it does not fully represent the physics behind the process, but helps to understand the separation that occurs, on a superficial level. This “naïve model” is based on dividing the flow through the DLD array into separate streams. The number of streams depends on the geometry of the DLD array (see Figure 3). The array is often made of rows of pillars that are shifted by a fraction 1/N of the row’s width (which equals to the diameter of the pillar and the gap between two adjacent pillars). Thus, every N rows, the position of the pillars is the same (see Figure 3), and the fluid flow is divided into N streams. This row shift of 1/N is also denoted as row shift ε. Each stream carries the same current of fluid. Since the flow speeds vary across the gap and in between the rows, the streams are not of the same width. Especially around the pillars the streams are especially narrow. If a particle or cell has a diameter bigger than these narrow streams, they are—in the “naïve model”—not able to follow the stream and migrate to the next stream, while a smaller particle can follow the streamline along the flow (see Figure 3a). In this way, bigger particles get “bumped” along the array towards the side (perpendicular to the flow), and separated from smaller ones, that can “zig-zag” between the pillars and follow the flow.

A more detailed discussion on more accurate models (including multiple critical diameters [104] and particle–particle interactions [105,106]) will be published soon by this author.

The shape of a cell or particle in a DLD array can have a massive impact on its hydrodynamic radius. The curved and elongated shape of trypanosomes and the biconcave shape of an RBC might inhibit a proper separation if the height of the DLD array is not carefully selected (see Figure 3b). Meanwhile, with an overly high array both cell types—despite being massively different—exhibit a very similar effective hydrodynamic diameter, simply because the trypanosomes can undulate freely (see Figure 3b). If the array is chosen to be very shallow (e.g., 4 µm), both cell types are forced to pass through the arrays while being vertically constricted. With an intermediate height, however, the larger trypanosomes are forced to undulate horizontally, while RBCs can align themselves vertically, resulting in a pronounced difference of their respective hydrodynamic radius.

NB: The complex motility of trypanosome has here been described as “undulating” for reasons of simplicity. A more detailed model of its motility has been discussed by Alizadehrad *et al.* in 2015 [107].

One of the disadvantages of DLD is that the arrays need to be tailored towards the cells which are to be separated and diagnosed on the chip. In addition, this tailoring is done by adapting the geometry of the DLD array: size and shape of the pillars, how far these pillars are apart, and the angle between the pillars and the main channel walls/the row shift ε. However, there have been several approaches to make DLD arrays “tunable” and thus adaptable over a larger range. One way to tune the sizes of separated cells is the combination of DEP and DLD [88,89,90]. Another way of tuning is combining DLD with non-Newtonian fluids, which change their viscosity depending of the flow velocity (and thus the shear forces the pillars exert on the liquid; shear-thinning) [108]. For general combinations of passive separation techniques (e.g., DLD) with active separation techniques (e.g., DEP, SAW, optical or magnetic) Yan *et al.* coined the phrase “hybrid microfluidics” [109]. 

For an overview of the pros and cons of DLD in the context of single cell separation and diagnostics, refer to Table 3 below.

Additionally, many DLD applications work based on a naïve and simplified model.

#### 2.1.3. Deformability-Based

A further technique for cell separation towards diagnostic chips that has been employed—both within and without DLD—is deformability-based separation. Not only are different cell types of different stiffness or elasticity (often measured in the Young’s modulus), but also the infection with parasites can alter the elasticity of cells. The most prominent example thereof is the increased stiffness of red blood cells infected with *Plasmodium falciparum* (iRBCs) compared to healthy red blood cells (RBCs) [110]. While this fact in itself has intriguing implications with sickle-cell anemia [110] and immunity towards Malaria, it has also been used to separate iRBCs from RBCs, both in theory [98,110,111] and practice [14,112]. One especially clever approach was presented by Guo *et al.* in 2016, where an oscillating flow separated rigid from elastic RBCs through an asymmetric filter array (see Figure 4).

In the same year, Park *et al.* improved this approach to separate white blood cells (WBCs), RBCs and CTCs from each other [113]. Wang and coworkers recently combined deformability-based separation with magnetic-based techniques to separate CTCs, RBCs, and WBCs from “liquid biopsy” samples [114]. In 2018, Hongmei Chen reported the separation of CTCs, RBCs and WBCs from spiked peripheral blood samples using a combination of deformability-based and inertial separation [115]. Zhou *et al.* predicted that it is possible to combine deformability-based and electrokinetic separation, which relies on different shear moduli instead of ratchets [116]. While their computations are based on inanimate particles, this could lead to continuous cell and particle separation in ratchet-free and clogging-resistant devices.

For an overview of the pros and cons of deformability-based separation assays in the context of single cell level diagnostics, kindly refer to Table 4.

#### 2.1.4. Margination and Dean-Flow

Furthermore, independently of a DLD array, iRBCs have been separated from healthy RBCs using a constriction within a microfluidic channel at high hematocrit values (=high concentration of RBCs in the sample, basically an only slightly diluted blood sample) [117]. This separation effect, in which some cells (e.g., iRBCs, leukocytes) are accumulated along the margins of a long channel has been called margination [117,118] and is a naturally occurring phenomenon in our smaller blood vessels [7]. It can be even enhanced by using a viscoelastic fluid as medium to create a high-throughput detection system [13]. Similarly, spiral channels have been used to separate different cells by creating a Dean flow. 

Xiang *et al.* recently even combined a Dean drag force-inducing spiral channel with a DLD array for a two-step separation of CTC from RBCs and white blood cells (WBCs) [119]. The underlying concept is that within curved channels two opposing forces are active on all particles and cells, the Inertial Lift force $F_{L}$ and the Dean drag Force $F_{D}$. 

The Inertial Lift force $F_{L}$ is composed of the shear-gradient-induced lift (caused by the parabolic flow profile inside the curved rectangular microchannel) and the wall effect (caused by the asymmetric wake of the particle near the wall), which push neutrally buoyant particle away from the center of the channel and the walls, respectively. This force $F_{L}$ can be calculated by [120,121,122,123,124]:(1)$$F_{L}=f_{L}\left(Re,\;x_{L}\right)\rho v_{max}^{2}\cdot 16r^{4}/D_{h}^{2}$$
where the lift coefficient $f_{L}$ is a function of the Reynolds number of the flow $Re\;\left(Re=\rho v_{max}D_{h}/\mu \right)$ and the particle position $x_{L}$ along the channel’s cross-section (in respect to the channel’s center). ρ and μ are the density and the dynamic viscosity of the fluid, respectively. $v_{max}$ denotes the maximal velocity within the microchannel, $D_{h}$ is the hydrodynamic diameter of the microchannel and shape and aspect ratio dependent (in first approximation: $2\cdot \frac{Width\cdot Height}{Width+Height}$), and r is the particle radius.

The Dean (drag) force $F_{D}$ on the other hand, will only form at relatively high Reynolds numbers in curved microchannels, due to the non-uniform inertia of the fluid in the inner and outer segments [124] of the channel. This Dean flow consists of two counter-rotating Dean vortices forming in the top and bottom halves of the channel (see Figure 5), and exerts additional transverse drag forces on particles [124]. 

This results in a displacement of particles flowing in a curved microchannel at high velocities. The flow in such a curved channel is defined by the dimensionless Dean number (*De*) [126]:(2)$$De=Re\sqrt{\frac{D_{h}}{2R}}$$
with *R* as the radius of the channel’s curvature. The maximum Dean drag force acting on a particle or cell in a curved channel can be estimated by Stokes drag force [125,126,127]:(3)$$F_{D}=6\pi \mu v_{max}r=1.08\cdot 10^{-3}\cdot \pi \mu r\cdot {\mathrm{De}}^{1.63}$$

As can be seen from Equations (1) and (3), these two forces scale very differently with the radius of the particle (linear vs. the power of four), hence it is possible to separate different cells (or other particles) by their size at higher flow velocities in a curved rectangular microfluidic channel. 

Further applications for Dean-flow-based microfluidics include the enrichment of human breast cancer cells from samples [128], human prostate epithelial tumor cells [129], or pathogenic bacteria from diluted blood samples [130]. To improve the cell separation prowess of Dean-flow-based applications, there have been some interesting recent improvements: through the addition of microstructures inside the channels, it is possible to separate blood cells using straight channels, which reduces the footprint of the designs and allows parallelization, as shown by Wu *et al.* in 2016 [131]. 

For an overview of the pros and cons of Dean-flow usage in the context of single cell separation and diagnostics, refer to Table 5.

#### 2.1.5. Surface Acoustic Waves (SAW)

Another technology that recently received increased attention in combination with microfluidic single cell diagnostics is called surface acoustic waves (SAW). SAW are generated by applying two conductive interdigital transducers (IDT) onto a piezo-electric carrier material. Interdigital here is rooted in Greek, and translates to “between fingers”, meaning that two transducers are shaped like combs that are pushed into each other, without actually touching (see Figure 6).

If a voltage is applied between these comb-shaped transducers, then the piezoelectric material in between is forced to dilate or contract. Applying an alternating voltage forces the piezoelectric material into oscillations, which then are called surface acoustic waves. These surface acoustic waves can either be standing (SSAW) or travelling (TSAW), depending on the design of the IDT and the frequencies of the alternating voltage. Now, a microfluidic channel can be superimposed onto this piezoelectric substrate and carry cells and particles perpendicularly to these surface acoustic waves (see Figure 7).

SSAW exert acoustic forces (*F_a_*) on these cells and particles proportional to their volume [132]: (4)$$F_{a}=-\left(\frac{\pi p_{0}^{2}V_{p}\beta_{m}}{2\lambda }\right)\phi \left(\beta ,\rho \right)\sin2kx$$
(5)$$\phi =\frac{5\rho_{p}-2\rho_{m}}{2\rho_{p}+\rho_{m}}-\frac{\beta_{p}}{\beta_{m}}$$
with *p*_0_ as the pressure amplitude caused by the SSAW, *V_p_* the volume of the particle, *β_m,p_* the compressibility of the medium and particle, respectively, *λ* for the ultrasonic wavelength, *ρ_m,p_* for the density of the medium and the particle, k for the wave vector and x for the distance to the pressure node.

Along the same axis as *F_a_*, but with opposite direction, there are viscous forces *F_v_*, which scale with the radius of the particle:(6)$$F_{\nu }=-6\pi \eta rv$$
with *η* for the medium viscosity, *r* for the particle radius and *v* the relative velocity of the particle with respect to the medium. Hence, acoustic forces dominate in larger particles but not in smaller particles, which allows the usage of SSAW to separate particles by size, if the other parameters are chosen accordingly. 

TSAW can also be used for particle separation, but the acoustic radiation forces on cells and particles are much lower compared to SSAW at the same frequency and power input [133]. While higher actuation frequencies can increase the effectiveness of TSAW, they also can disturb the laminar flow within the microfluidic device by causing acoustic streaming [134,135]. This in itself, however, has given rise to other useful applications of TSAW [133,134,135], including pumping [136] and processing of multicellular organisms (e.g., *Caenorhabditis elegans*) [137]. 

A set of four IDTs, arranged along the sides of a square, can be used to create so called acoustic tweezers, where the interferences of the four IDTs can create trapping nodes that can hold individual cells. By changing the frequencies and amplitudes of the SAWs emitted by the IDTs, the trapped cells can be moved within the volume of the acoustic tweezers [138]. Similar to SAW in general, there can be three types of acoustic tweezers: (1) Standing-wave tweezers, (2) Traveling-wave tweezers, and (3) Acoustic-streaming tweezers [138].

For an overview of the pros and cons of SAW technology in the context of single cell separation and diagnostics, please refer to Table 6 below.

#### 2.1.6. Optical Trap and Tweezers

Light can interact with matter on the microscopic scale, as proven by Arthur Ashkin almost five decades ago [139]. The operating principle of optical tweezers is rather simple (see Figure 8), as demonstrated on laser light (orange) and a transparent spherical particle. A normal laser beam has a roughly Gaussian intensity profile (see orange intensity profile top left in Figure 8) and parallel rays of photons. If these rays enter a spherical particle of higher refractive index (e.g., a glass microsphere) the photons get refracted and in turn exert a force onto the sphere. Since there are more photons per interval where the intensity is higher, the sphere will pulled towards the center (by the gradient force $F_{gradient}$, see Figure 8) and pushed along the propagation of the laser beam (by the force of the scattered photons $F_{scatter}$).

If an additional lens is introduced into the light path, a position is created where the sphere can be held stably, since all the forces acting onto the sphere will be in equilibrium (see Figure 8 right). If the particle moves out of this point, the sum of the resulting forces will pull the particle right back into this stable position. Additionally, if the beam is moved away, the particle will be forced to follow. This is the basic on which optical trapping operates.

Since many cells have a higher refractive index (or are optically denser) than water or their culture media, they can be manipulated with optical traps, akin to the particle in Figure 8. Objects that have a lower refractive index than the surrounding medium (e.g., air bubbles), will get pushed away from the beam.

For alternative setups of optical traps, it is also possible to use two counter-propagating beams [140,141], or two parallel lasers to generate interference to trap particles [142], or two inclined fiber-coupled laser beams, which create a stable trapping position below the intersections [143].

Another way to generate optical traps is by shining an expanded laser beam onto a spatial light modulator (SLM) [144,145,146,147,148,149,150]. The resulting optical traps are generally referred to as “holographic optical traps (HOTs)” and are based on the interferences of many parallel phase shifts, all caused by being reflected off the SLM. The advantage of HOTs is that with one setup it is easy to generate dozens of traps, and move them in all 3 dimensions. An intermittent problem—the emergence of unwanted additional HOTs (so called ghost traps [151]) has in the meantime been reduced by the introducing small disorder [152], or random phase elements [153].

The application of optical tweezers for the handling of single cells and for diagnostics on the single cell level has recently been discussed a lot [138,141,154,155]. However, it can as yet not be used as the only technique for any kind of diagnostics, and is either combined with other tests to create assays [63,66], or needs massive automation and robotics [156] or artificial intelligence to arrive at a diagnosis [156,157]. The usage of optical traps for single cell diagnosis is mostly limited to optical stretching [158], cell-sorting [66,154,159,160,161,162], and measuring of refractive indices [140,162,163,164]. Especially in combination with automatization, optical traps for single cell diagnostics have a great potential, but currently it is among the more expensive and less-performing technologies. 

The pros and cons of optical traps in the context of single cell separation, analysis and diagnostics are listed in Table 7 below.

### 2.2. Combined Separation and Analysis on Chip

#### 2.2.1. Droplet Microfluidics

All the technologies discussed so far have two things in common: they are all only used to separate different kinds of cells; and they all are used in continuous phase microfluidics, which means that one stream of liquid contains all the cells and particles. In contrast, there exist several techniques for which the cells are encapsulated in individual droplets (mostly of water or aqueous solutions) that are separated by another phase, either oil or even air. This allows not only to encapsulate single cells within individual droplets, but also to analyze their secretions, metabolites and (after cell lysis) their contents. Kaminski and Garstecki published a comprehensive overview on those techniques in 2017 [165]. In brief, there are the following techniques:

Droplet microfluidics: Any emulsion system of water droplets in an oil carrier phase (or water-droplets-in-oil-droplets-in-water; or any permutation thereof) can be used to separate individual cells or solutions from each other. This is generally done with an x-intersection on a microfluidic chip, where the aqueous phase is pinched off into droplets by two streams of oil (see Figure 9a) Using any kind of forces (shear, drag, coulomb, inertial, etc.), droplets can be merged to combine their contents and to start reactions towards a diagnostic signal (see Figure 9c), like for example an exosome immunoassay for cancer diagnosis [166]. 

Individual droplets can be handled, for example, using acoustophoresis/SAW [134,167], hydrodynamic or microfluidic changes by geometries [168,169,170,171], elective emulsion separation [172], in-line passive filters [173], or even DLD [99]. It is also possible to manipulate the contents inside these droplets using DEP [174,175], magnetic fields [176,177], or beads and microfluidic ratchets [178].

Different from that is digital microfluidics (DMF), which employs the electro-wetting effect. Using an external electric field, the interface energy between the polar (water) droplet and the (dielectric) surface can be locally modulated, and thus the contact angle between the droplet and the surface can be reduced. This effectively renders the surface locally more hydrophilic and lets the droplet migrate along this hydrophilicity gradient. Individual droplets can be moved and directed like this, separated, merged or stored as illustrated in Figure 9b. One commercial DMF platform for diagnostics in newborns and children has just been presented [179].

As a general note: droplet microfluidics can be used in many instances like traditional well-plate assays, with the few following adjustments: Well plates are static and highly parallelizable, while droplet microfluidics are more dynamic (i.e., the well plates rest on their plates, while the droplets generally have to move in sequence to the next operational point) and can be handled in high-throughput series. However, especially in drug assays, (droplet) microfluidics-based assays have a great potential to surpass well-plate-based approaches [63].

For an overview of the pros and cons of droplet microfluidics for single cell separation, analysis and diagnostics, please refer to Table 8 below.

#### 2.2.2. Optical Density/ Refractive Index

Using the refractive index of a single cell within a microchip for potential single cell diagnostics was reported over a decade ago, e.g., by Liang *et al.* [180]. Over the past 15 years, several applications have been reported, based on advanced measurements of refractive indices of single cells. The main measure of their precision is the Refractive Index Unit (RIU). The refractive index is the ratio of how fast light travels in vacuum to how fast light travels in a medium (e.g., water, cytoplasm). Proteins and DNA have a higher refractive index than water, and thus the local refractive index of a cell can be used to measure the local concentration of protein at any volume of a cell. The refractive index itself has no unit or dimension. The RIU can be seen as the (smallest) portion of (local) change in the refractive index that can be measured by any given method.

Exemplary methods for measuring the refractive indices of single cells onboard of microfluidic chips include light scattering [140], phase contrast microcopy [181], laser resonant cavity [180], Fabry-Pérot cavity [182], Mach-Zehnder interferometry [183], or combining an optical trap with a grating resonant cavity [184].

To generate a combined separation and analysis on-chip device, any of the above-mentioned separation approaches can be combined with any measurement of the refractive index of the isolated cells. The refractive index of a cell is a key biophysical parameter and correlates to biophysical properties including mechanical, optical and electrical properties [185]. With the combination of microfluidics, photonic and imaging technologies, it is now possible to study the 3D refractive index of a cell in the sub-micron regime, as Liu *et al.* detailed in their review in 2016 [185]. In addition to this, it is possible to use Brillouin microscopy, where the optical phase shift of a cell gives information about the cells local stiffness and water content, given the local refractive indices are known [186], or measured alongside using one of the techniques listed in Table 1. Three major approaches for probing the cell refractive index can be summarized, as shown in Table 9, below:

For a more in-depth discussion on these techniques, see the review by Liu *et al.* in [185]. A general overview of the pros and cons of these techniques can be found in Table 10 below.

#### 2.2.3. Paper Microfluidics

Paper microfluidics (PMF) was first used for portable diagnostics in 2007 [200] as a low-cost alternative to continuous phase or droplet microfluidics. Instead of using walls and hollow structures, paper microfluidics uses paper as a hydrophilic stationary phase (that carries the fluid) and a hydrophobic phase (paper treated with wax, photoresist, graphene or other substances) to block the fluid. Since paper is, on the microscopic scale, a tangle of fibers, most cells and similarly big particles are retained, while the smaller and soluble parts travel along the paper as they do in thin layer chromatography (see Figure 10). Thus, PMF is used more often to analyze the contents of lysed cells (e.g., DNA [11,16,20]) than entire cells. However, it also possible to have a PMF device that starts out with a sample of cells and media, first separates the cells, then lyses the isolated cells and runs molecular analysis on them. One example is a field-applicable test using foldable paper slips for diagnosis of Malaria in infected blood samples [11].

The adaptability and compatibility of PMF with smart phone analysis, the mass-producibility and readily available and cheap materials, along with the possibility to make three-dimensional devices by folding, cutting, and washing the paper allowed paper microfluidics to make enormous advances in the last 12 years. From sample collection to signaling the result, paper-based assays have been used for every step along diagnostic pipeline, generating “a formidable toolbox of advanced strategies for fluid and analyte manipulation in paper-based assays” [33]. With the appropriate (pre-)treatments, paper-based assays allow portable and hand-held versions of analytical techniques, such as DNA separation and nucleic acid amplification, outside of specialized laboratories, even unlocking them for field tests in low- and middle-income countries. 

While paper-based microfluidic assays are a cheap and reliable alternative to other single cell level diagnostic devices, they need a different approach than other microfluidic techniques and also face specific challenges especially along their translation and commercialization: (1) Integrating the multiple tasks into one single system and (2) obtaining clinical validation, while (3) adhering to the various EU/US/national regulations, (4) up-scaling of production, especially with multiple pre-treatments on the paper, and (5) maintaining storability. All this notwithstanding, PMF is arguably the leading approach to single cell diagnostic chips, as shown by many reviews over the last two decades detailing the usages of PMF diagnostic chips, also called microfluidic analytic devices (µPADs) [20,33,78,201,202,203,204,205]. PMF has been combined with many other techniques to create a plethora of applications: from combining PMF with immunoassays [206] for Hepatitis C-tests, via on-board batteries [207] or cell phones [203] to generate field-applicable µPADs, to proteomics [208] parasite diagnostics. How established µPADs are in our everyday life is demonstrated by the usage of paper microfluidics for cheap home tests for male (and female) fertility and sperm DNA integrity [16,17,18,19,20]. 

The main difference in cell handling in PMF compared to continuous phase or droplet microfluidics is that cells generally do not travel through or along the paper. Individual molecules (e.g., after the cell was lysed), however, can diffuse through the paper in a fashion very similar to (thin layer) chromatography. This also opens up targeted retention of cells during washing steps (e.g., [11]).

For an overview of the pros and cons of PMF in the context of single cell separation and diagnostics, please refer to Table 11 below.

### 2.3. Molecular Analysis of Single Cells

If not only the cell as a whole, but also subcellular and molecular parts of the cell, are the target of a diagnostic chip, it is a standard step to lyse the cells, wash, filter, optionally amplify selected parts and finally verify the existence and concentration of the molecules of interest. For this molecular analysis of single cells, the following technologies are most commonly used in today’s diagnostic chips:

#### 2.3.1. Polymerase Chain Reaction (PCR, Nested PCR, qPCR/ RT-PCR)

Especially after lysing cells to check for intracellular or subcellular markers (e.g., DNA, RNA fragments, antibodies), the sheer plethora of compounds can be hard to read out when looking for a single structure. In this case, an amplification step is introduced to replicate the specific parts of this wild mixture of substances. The most basic technique, polymerase chain reaction (PCR) simply emulates a process that happens within us every day (see Figure 11). To verify the presence of a suspected target (e.g., the Malaria causing parasite *Plasmodium falciparum*), all the DNA in the cell is denaturized (basically unfolded from the double helix). Since the DNA of individual life forms (e.g., *P. falciparum*) have distinctive already-known sequences, we can select these sequences to be multiplied by adding primers, consisting of the corresponding base pairs (see Figure 11).

Using an enzymatic cleaving of the double stranded DNA into single strands allows individual nucleotides to assemble on these single strands and form two new double stranded DNA molecules. While our bodies can perform this at body temperature, on a chip, this amplification loop is carried out at higher temperatures: cleaving of double stranded DNA (Denaturation) at 94–96 °C; recombination of single strands with matching primers (Annealing) at ca. 68 °C; and the adding of further bases (Elongation) at ca. 72 °C. Going through this amplification loop (see Figure 11 (1–3)) once doubles the amount of targeted DNA. However, with each copy, the strand becomes shorter by a basis. This happens both in vitro and in our bodies and is believed to be a main factor of aging [209]. 

Within the confines of a microfluidic chip, such different temperatures can best be achieved by having a long channel meandering over different heating stages, as demonstrated exemplarily by Ma *et al.* in 2019 [210]. This can also be done with droplets that already contain the primers and nucleotides [210]. The application of PCR is especially widely applied for diagnostic chips for malaria detection [11,13,211] and other parasites [43,212].

With our growing understanding of the biophysics and biochemistry of life, the PCR technique has been advanced. Nowadays, it is commonplace to already obtain real-time quantified results (real-time PCR; RT-PCR or quantitative PCR, qPCR) in between the amplification steps. Basically, this is done by adding a fluorophore to the DNA-primers (or other prominent parts), which emits a fluorescent signal only when bound to a strand of DNA (or which stops emitting after being bound to other nucleotides). By comparing the intensity of fluorescent signal after each amplification loop, it is possible to quantify the amount of target DNA within the sample. The incorporation of qPCR into single cell diagnostic chips has recently been discussed more in detail by Reece *et al.* [37]. In summary, nowadays, microfluidic single cell separations based on optical manipulation, microfluidic large-scale integration, hydrodynamic cell sorting/stretching, and droplet microfluidics, have achieved the high-throughput capacities necessary to allow—in combination with PCR-based analysis—population-wide screenings of samples on the single cellular level. 

To reduce non-specific binding in PCR products that arise from unintended primer binding sites, one loop of PCR can be nested inside another, using a different set of primers. This nested-PCR is also a widespread practice in (single cell) diagnostic chips and its application has recently been reviewed and discussed with other PCR techniques [213].

A combined overview of the pros and cons of both PCR and LAMP-based techniques can be found in Table 12 below.

#### 2.3.2. Loop-Mediated Isothermal Amplification (LAMP, NAAT, LAMPport, NINA-LAMP)

An alternative technique for diagnosing sub-cellular targets is loop-mediated isothermal amplification (LAMP), which can run at a single temperature of around 65 °C and inside a single tube where the cell lysate is mixed with DNA polymerase and a set of four (or more) specifically designed primers [214]. The fact that it needs less equipment to be run, and that it can even be more portable [55], or fully non-instrumented [48], makes LAMP and its derivatives very promising for use in the field for resource-scarce settings [11,22,47,48,55,61]. 

One example for a field-applicable LAMP device is a chip for nucleic acid amplification test (NAAT, see Figure 12). In brief, operating the NAAT consists of the following steps: First, the cells are lysed and the lysate is introduced into the chip. Second, a buffer is used to wash the lysate and stabilize the targeted molecules. Third, water is added to the solution, which may already contain reagents and enzymes. Fourth, by heating the reaction volume to 65 °C, the amplification loop is started. In some cases, the heater also melts away an encapsulation between the (lyophilized) enzymes and reagents and the water [61]. Finally, after the amplification has run for the desired amount of time, the reaction mixture is excited with a light source and the fluorescent response of the reaction mixture is measured to assess the presence and concentration of the target [61]. 

Even more advanced and exciting are LAMP-derived techniques, specifically designed for the deployment in areas with very limited access to resources, e.g., field work in endemic areas of parasitic diseases like Malaria, LAMPport (portable) [55] and non-instrumented nucleic acid amplification LAMP (NINA-LAMP) [48]. With further tweaking, it is possible to attain field-applicable multi-parasite or multi-disease tests at low costs—if it can be combined with a proper separation technique, like DLD or paper microfluidics. 

#### 2.3.3. Proteomics, Metabolomics, Transcriptomics and Polyomics

A very important set of techniques for diagnostic chips is the group of prote-/metabol-/transcript-/gen- or poly ‘omics’. According to Haring and Wallaschofksi, “omic-metrics including the Phenome (physical traits such as body height, weight, or specific personality characteristics), Metabolome (complete set of small-molecule metabolites to be found within a bio- logical sample), Proteome (entire set of proteins expressed by a genome, cell, tissue, or organism), Transcriptome (information about the expression of individual genes at the messenger ribonucleic acid level), Genome (complete set of genes in the […] organism)” [215]. Single cell metabolomics is considered a crucial element for targeted drug discovery [216], and already ten years ago, single cell analysis was the new frontier in ‘omics’ for Wang and Bodovitz [217].

In general, all single cellular ‘omics’ share a basic setup: a target cell (or several) is lysed, their contents may be treated and are chromatographically separated and analyzed using mass-spectroscopy (or similar techniques). Most often, separation and detection is done using gas chromatography with mass spectroscopy (GC-MS) [218], high-pressure liquid chromatography and mass spectroscopy (HPLC-MS) [219], and capillary electrophoresis with mass spectroscopy (CE-MS) [220]. Since a single cell contains a plethora of substances that can be found after the chromatography in the spectra, it is usually not possible to identify individual peaks, but rather “fingerprints” of overlying bands. By specifically adding individual molecules to this cellular cocktail, changes in the overlying bands can be assigned individual substances in a given setting. With careful sample treatment, -omics can be very well used for diagnostic chips and to unravel to exact roles of individual substances within these cells, their metabolism, their transcription during cell division, or all of the above. Proteomics has been massively employed to elucidate the host interactions of SARS-CoV-2, the causative agent of COVID-19 [27,28,29,30,31,32], the greatest pandemic in our millennium.

For an overview of the pros and cons of employing polyomics towards single cell level diagnostics, please refer to Table 13 below.

## 3. Implementation

### 3.1. Point-of-Care Diagnostics (POC)

Point-of-Care (POC) means that these diagnostic devices can be used at the patient level and deliver a diagnosis locally and quickly—without need for taking samples, sending them away for analysis and having to continue care while waiting on the outcome of the diagnostic test. This is often achieved with the LAMP test and its derivatives, LAMPport and NINA-LAMP [22,47,48,55,61]. However, many other, especially microfluidic, approaches have also been tested and discussed for their aptitude to be used for POC diagnostic tools [38,45,205,212,221,222,223,224,225].

### 3.2. Biosensors in the Developping World and the Need for ASSURED

Many publications, which demonstrate microfluidic test devices mention at least some of the WHO’s “ASSURED” criteria (i.e., Affordable, Sensitive, Specific, User-friendly, Rapid and robust, Equipment-free, Delivered to the users who need them) for low-cost sensors for the developing world. While the functionality “ASSR” of the low-cost sensors is often easily achieved, their ready user acceptance and field-applicability “UED” often lags behind [201]. 

Microfluidics devices make it possible to carry out complex analysis outside of highly equipped laboratories, often in a hand-held, portable, and field-applicable fashion [226], at least that is the promise of the “lab on a chip” idea. In reality, for many steps along the way, we experts experience it to be a “chip in a lab” first, with many hurdles to overcome, including the “valley of death”: the gap between the academic research and industrial/real-world application. 

## 4. Discussion

Given the steady increase in publications and research carried out towards the creation and usage of single cell diagnostic chips, it can really be regarded as a hot topic within the microfluidics community. On the diagnostic side, single cell diagnostics means that infections and diseases can already be diagnosed with a minimal sample volume and from a single pathogenic cell. On the production side, this means that high-resolution techniques have to be used (e.g., soft lithography), but also that it is possible to parallelize and mass produce these microfluidic chips. 

The vast adaptability of microfluidics to other techniques and technologies to handle and characterize cells, from bulk to individual, opens new possibilities and synergies every year. While this plethora of opportunities continues to be explored, it is always a rather long stretch from the “chip in a lab” to the field-applicable “lab on a chip”. Many great diagnostic chips that could help study, diagnose and eventually eradicate diseases—especially neglected tropical diseases (NTDs), do not make it to the final deployment in the field. It is a sad state of affairs and a symptom of some mismatches in the research and funding landscape, that many projects aiming to understand the fundamentals are funded, and even more projects are funded to develop actual applications and physical single cell diagnostic chips, but that only very few devices ever make it across this “valley of death” to their field application. Several funding bodies like the Bill and Melinda Gates Foundation or the UK’s Grand Challenges Research Fund support research to develop such chips for low- and middle-income countries. However, the—now—most important step to get the chips mass produced, certified and deployed, is sadly virtually unfunded; an immense potential to actually make a positive impact on the world is not being seized. 

## 5. Conclusions

In conclusion, there have been many technologies that can be combined to create powerful single cell diagnostic chips, but without a massive change in the system, many of these chips will remain “chips in a lab” rather than unleashing their true potential as “labs on a chip” at the point of care.

## Figures and Tables

**Figure 1 micromachines-11-00468-f001:**
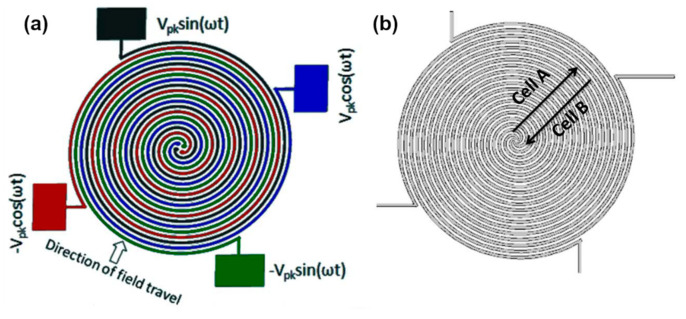
Four-arm spiral quadrupole electrode used by Menachery *et al.* [54]. (**a**) Schematic of the four-arm spiral microelectrode array comprising four parallel spiral elements of 30 mm in width and spacing. The electrodes are energized with a 90° phase shift with respect to each other. (**b**) Working principle of the chip. While cell type A (e.g., red blood cells) is expelled from the electrode array, cell type B (e.g., trypanosomes) is concentrated into the center of the array. Both processes take place simultaneously. Reproduced with permission from [54].

**Figure 2 micromachines-11-00468-f002:**
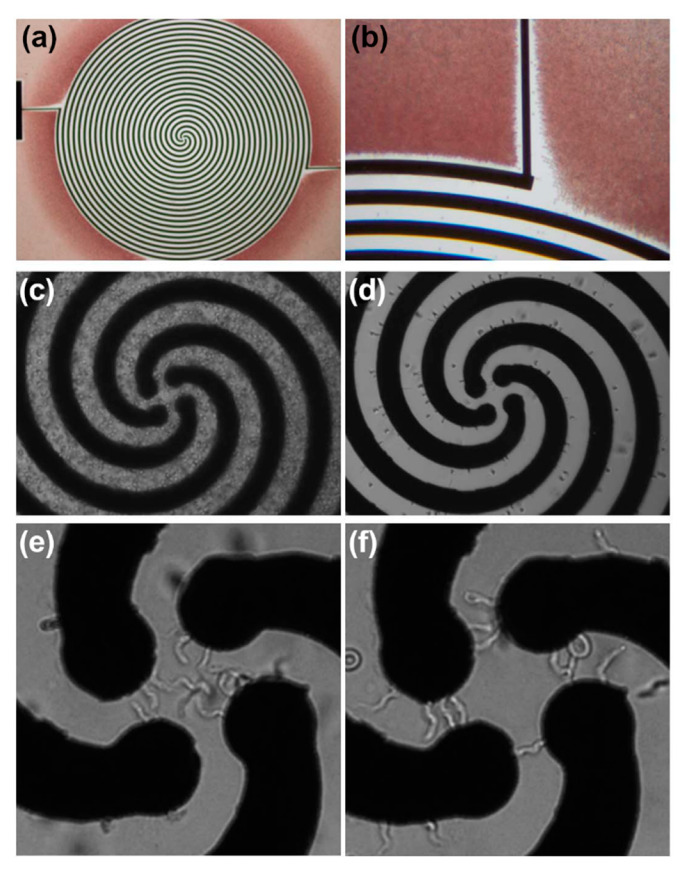
Enrichment of trypanosomes from infected blood. Total width of the spiral array is 2.9 mm, electrode width and spacing is 30 mm. (**a**,**b**) Micrograph following a separation process, with the RBCs having been pushed away from the electrode array. (**c**) Parasitized blood on the spiral electrode array. (**d**) Mouse RBCs are levitated and carried to the outer edges of the spiral. (**e**) Trypanosomes accumulate in the center of the spiral and undergo circular translational motion. (**f**) Trypanosomes are trapped along the electrode edges in the center of the spiral upon switching the AC voltage from quadrature-phase to an opposing two-phase. Reproduced with permission from [54].

**Figure 3 micromachines-11-00468-f003:**
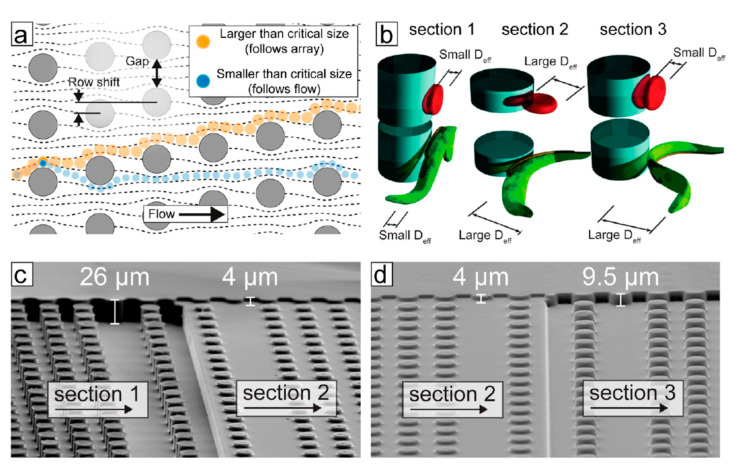
The principle the “naïve model” of deterministic lateral displacement, on the example of trypanosomes and red blood cells. (**a**) An array of posts divides a fluid-flow into many well-defined streams. Particles smaller than the critical size follow the streams, whereas larger particles follow a trajectory defined by the geometry of the array. (**b**) The effective size of particles is a function of their shape and orientation as they flow through the device. Device depth can be used to control the orientation of blood cells and parasites to maximize differences in effective sizes. (**c**,**d**) Scanning electron microscopy images of a poly(dimethylsiloxane) (PDMS) device, designed to separate trypanosomes from blood cells; the different sections achieve different separation steps. Adapted with permission from [8].

**Figure 4 micromachines-11-00468-f004:**
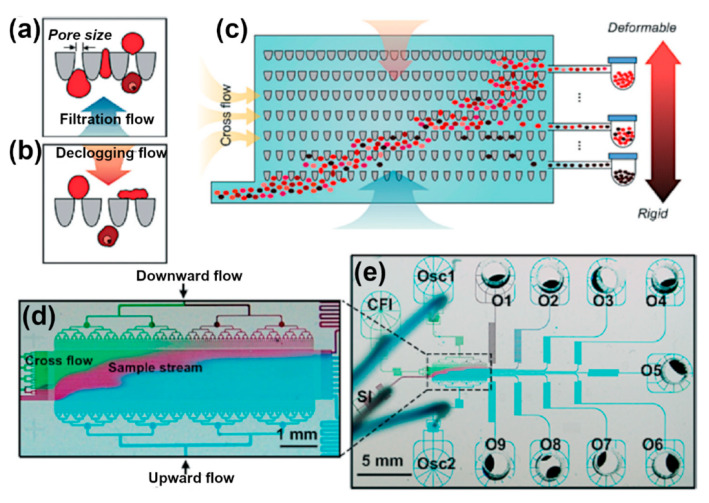
Design of the ratchet-sorting device. (**a**,**b**) Tapered funnel constriction allowing unidirectional flow of cells under oscillation excitation which consists of (**a**) upward filtration flow and (**b**) downward de-clogging flow; (**c**) cell sorting using a matrix of funnel constrictions. The cell sample is introduced through the sample inlet (SI) and forms a diagonal trajectory under the combined forces of cross-flow inlet (CFI) and biased-oscillation flows including oscillation inlet 1 (Osc1) for de-clogging and oscillation inlet 2 (Osc2) for filtration. More deformable cells, such as RBCs, will travel further up the matrix of funnel constrictions than less deformable cells, such as iRBCs, which will be blocked midways and be separated from the main population. (**d**) Image of microfluidic ratchet device infused with different food color dyes illustrating the diagonal trajectory of the SI through the ratchet-sorting device constituting a deformability gradient. (**e**) Image of the overview design of the ratchet sorting device as well the nine outlets (O1–9). Adapted with permission from [112].

**Figure 5 micromachines-11-00468-f005:**
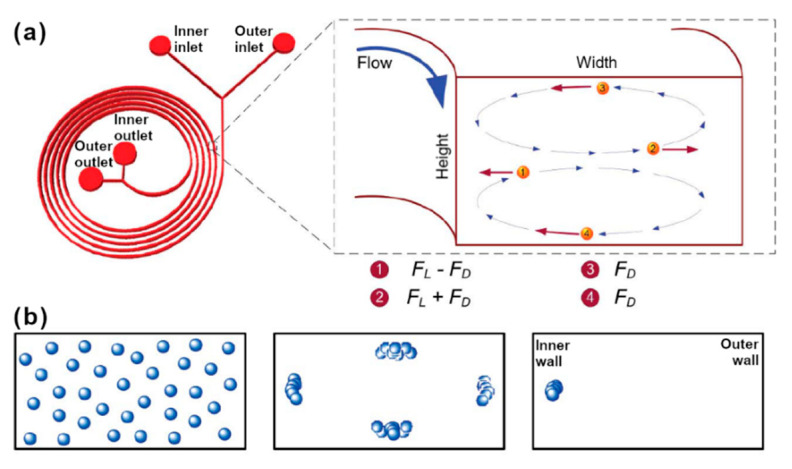
(**a**) Schematic of a spiral micro-particle separator. The design consists of two inlets and two outlets, with the sample being introduced through the inner inlet. Neutrally buoyant particles experience Lift forces ($F_{L}$) and Dean drag ($F_{D}$), which results in differential particle migration within the microchannel. (**b**) Microchannel cross-sections illustrating the principle of inertial migration for particles with $r/D_{h}∼\;0.05$. The randomly dispersed particles align in the four equilibrium positions within the microchannel where the Lift forces balance each other. Additional forces due to the Dean vortices reduce the four equilibrium positions to just one near the inner microchannel wall. Adapted with permission from [125].

**Figure 6 micromachines-11-00468-f006:**
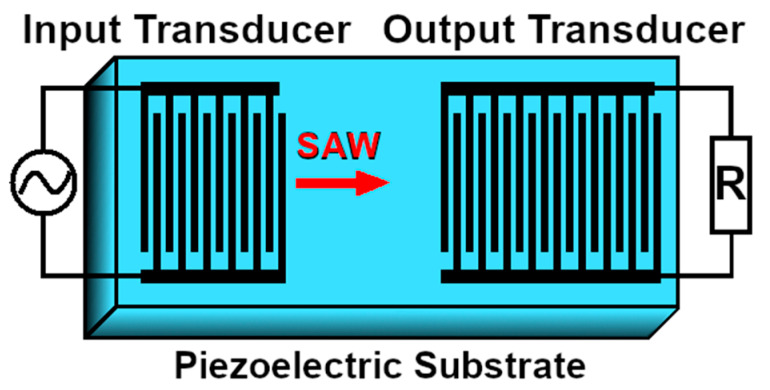
Interdigital transducers (IDT) (black) used to generate surface acoustic waves (SAW) on a piezoelectric carrier material. The input transducer generates mechanical forces from electrical voltage, while the output transducer can re-convert the mechanical oscillations into an alternating voltage.

**Figure 7 micromachines-11-00468-f007:**
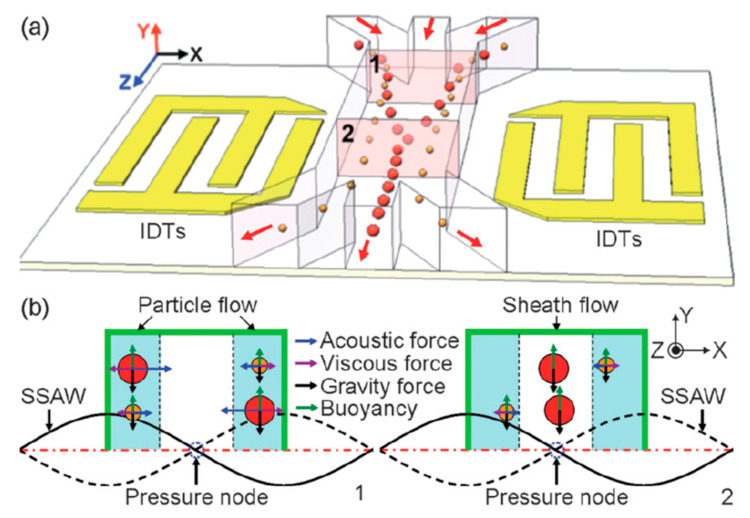
(**a**) Schematic of the separation mechanism showing particles beginning to translate from the sidewall to the center of the channel due to axial acoustic forces applied to the particles when they enter the working region of the SSAW (site 1). The differing acoustic forces cause differing displacements, repositioning larger particles closer to the channel center and smaller particles farther from the center (site 2). (**b**) Comparison of forces (normally in the pN range) acting on particles at site 1 and site 2, respectively. Reproduced with permission [132].

**Figure 8 micromachines-11-00468-f008:**
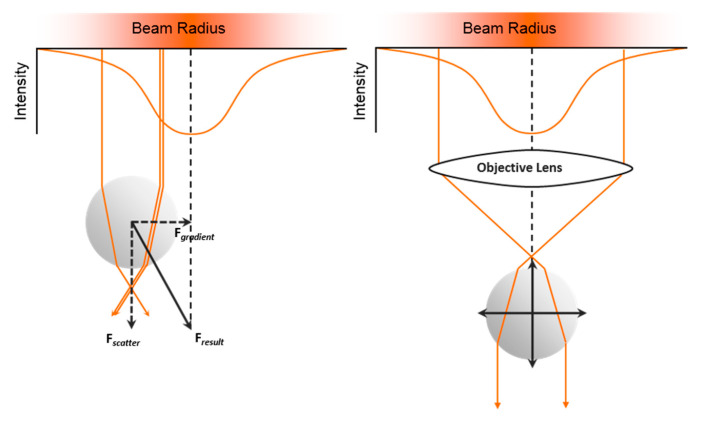
Operating principle of an optical trap/optical tweezers, demonstrated on laser light (orange) and a transparent spherical particle. (**Left**) A normal laser beam has a roughly Gaussian intensity profile (see orange intensity profile top left) and parallel rays of photons. Spherical particles of higher refractive index (e.g., a glass microsphere) refract photons that pass through them. By the law of impulse conservation, a force is exerted on the sphere (the scatter force *F*_scatter_), which propels the particle along the propagation direction of the laser. The higher intensity of photons at the center of the beam, results in a net force towards the center of the beam (by the gradient force *F*_gradient_). (**Right**) The introduction of an additional lens into the light path, creates stable trapping position. Here all optical forces acting onto the sphere are in equilibrium. Dislocating the particle from this point, results in a force that will pull the particle right back into equilibrium position. Image adapted from a sketch by Dr Eric Stellamanns.

**Figure 9 micromachines-11-00468-f009:**
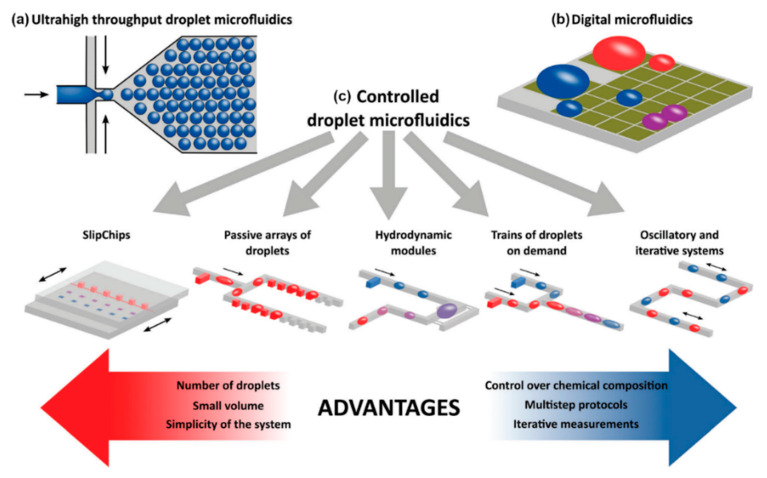
Schematic classification of droplet microfluidics that comprises three main groups of technologies: (**a**) ultrahigh throughput microfluidics characterized by the largest number of droplet bioreactors, (**b**) digital microfluidics that enables individual control over each droplet bioreactor, and (**c**) controlled droplet microfluidics which is a group of technologies that exhibit moderate throughput with the capability to address droplets in series. Adapted with permission from [165].

**Figure 10 micromachines-11-00468-f010:**
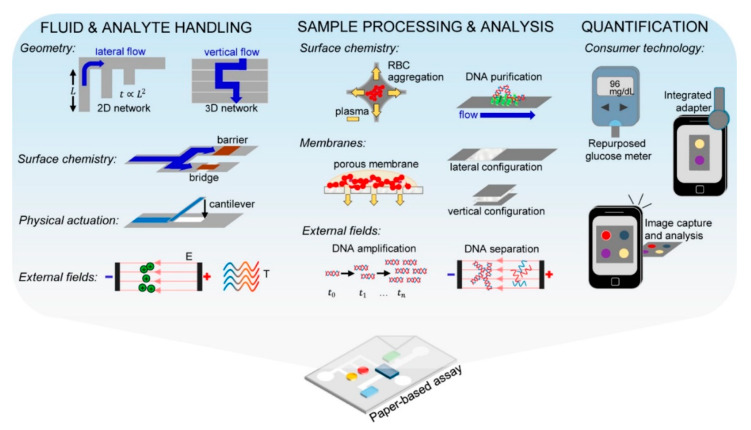
Paper microfluidic applications for diagnostics, including fluid and analyte handling, sample processing and analysis, as well as quantification. Reproduced with permission from [33].

**Figure 11 micromachines-11-00468-f011:**
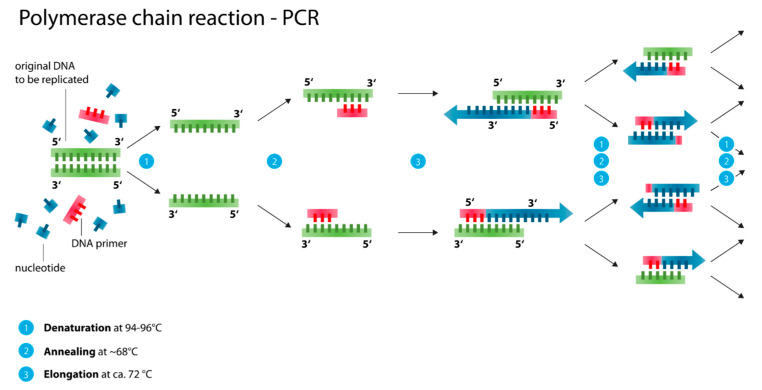
Scheme of the polymerase chain reaction: (**1**) Denaturation: The original DNA double strand is split into two complementary strands. While in our bodies this happens enzymatically at our body temperature, in microfluidic chips the DNA has to be heated to about 95 °C. (**2**) Annealing: Compatible DNA primers attach to the individual DNA single strands. These normally consist of several bases to increase the chances that this primer is specific and attaches as close to the end of the single DNA strands as possible. On chip this is done around 65–68 °C (**3**) Elongation: Beginning from the 3′-end of the DNA primer, individual nucleotides (bases) attach complementary to the respective single stranded DNA. Thereby, Arginine and Tyrosine complement each other and Cytosine and Guanine respectively. Elongation on chip usually takes place at around 72 °C. Images were adapted from Wikipedia user Enzoklop in 2014.

**Figure 12 micromachines-11-00468-f012:**
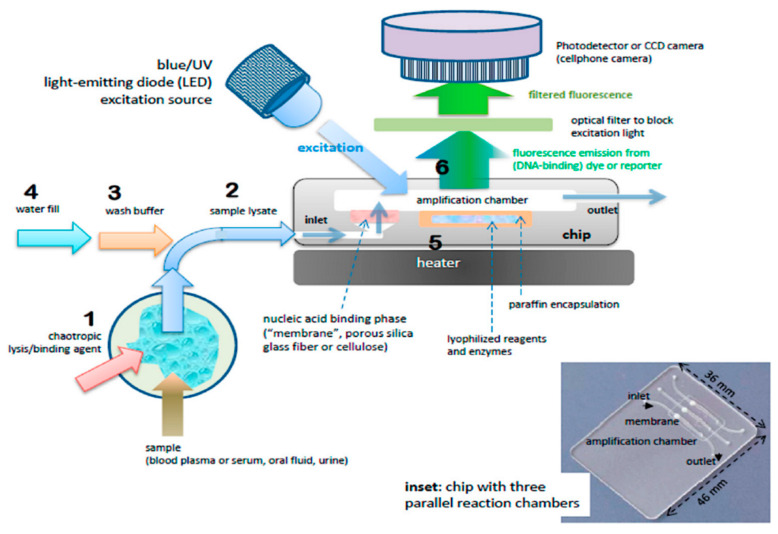
Process steps and (cross-section) schematic of chip for nucleic acid amplification test (NAAT). Chip has one or more flow-through chambers for isothermal amplification and includes a filter-like, flow-porous nucleic acid binding phase (e.g., silica glass fiber or cellulose) and is pre-loaded with paraffin-encapsulated amplification reagents (lyophilized polymerase, primers, fluorescence reporter DNA-intercalating, dyes, and other components). Operational steps: (**1**) sample is mixed off-chip with lysis/binding reagent buffer (containing e.g., chaotropic agent such as guanidinium HCl) that lyses virus and cells and promotes nucleic acid adsorption to binding media, e.g., silica glass fiber or cellulose (‘membrane’), (**2**) sample (~100 μL) is injected into chip with pipette or syringe, (**3**) ethanol-based, high-salt buffer (~100 μL) is injected into chip to wash the membrane (keeping most of the captured nucleic acid adsorbed to the membrane, (**4**) chamber (25 to 50 μL volume) is filled with water and sealed with tape, (**5**) chip is heated to amplification temperature (~65 °C) using a small (~1 Watt) electric-heater. The heating melts the paraffin encapsulation, releasing and reconstituting the reagents, (**6**) the amplification reaction is excited with a blue or UV LED, such that the DNA intercalating dye generates a fluorescence signal proportional to the amount of DNA amplicon produced. The fluorescence is measured by filtering the excitation light and detection with a photodetector of CCD camera, such as provided by a mounted cellphone. Reproduced with permission from [61].

**Table 1 micromachines-11-00468-t001:** Techniques applied to achieve single cell diagnostic chips.

Technique (Abbreviation)	Applications
Dielectrophoresis (DEP)	Separation of blood, infected blood cells, parasites
Deterministic lateral displacement (DLD)	Separation of blood, infected blood cells, parasites, CTC, spores, DNA, viruses
Deformability-based separation	Separation of blood, infected blood cells, parasites
Margination & Dean Flow	Separation of blood, infected blood cells, parasites, CTC
Surface acoustic waves (SAW)	Separation of blood, parasites, CTC, multicellular organism
Optical tweezers (OT)	Separation of rare cells, parasites, CTC
Optical density/refractive index	Fast optical analysis of cell size, shape and optic density
Droplet microfluidics	Separation and storage of cells, mixing with targeted start of reactions, post-lysis analysis of cell contents
Paper microfluidics	Separation of cells, post-lysis analysis of cell contents
PCR-based techniques	Targeted enrichment of subcellular fragments to verify their presence or identity of the previously separated cells
LAMP-based techniques	Similar to PCR, but generally also employable in environments with very limited resources
Prote-/metabol-/transcript-/poly omics	Gather data on all the parts of a cell, their transcriptomes or their metabolites respectively. Elucidating contents, pathways and interactions, (e.g., host–viral interactions of SARS-CoV-2).

**Table 2 micromachines-11-00468-t002:** Pros and cons of dielectrophoresis (DEP).

Pros	Cons
Adaptable to different cells by frequency	Needs electricity
Potentially parallelizable	Low throughput, unless 3D- electrodes are used
Compatible with other techniques (e.g., DLD, SAW molecular analysis after cell lysis)	Cell frequencies need to be experimentally found
-	Separation is rather slow and hard to automate

**Table 3 micromachines-11-00468-t003:** Pros and cons of deterministic lateral displacement (DLD).

Pros	Cons
Needs no electricity (e.g., using handheld syringes)	Prone to clogging
Tiny defects can be tolerated by redundancy	Trapped air bubbles can completely ruin separation
Can be parallelized	Low throughput due to tiny volume
Can be run constantly and separated cells can be collected for further processing in closed loop systems.	The separation process of DLD has not been fully understood so far and entails many different aspects that are thus far neglected.

Note: while the geometry of the array needs to be tailored to the cells that are to be separated in bulk samples, several techniques have been discovered that can tune the effective separation diameter to adapt an existing array to new cell types.

**Table 4 micromachines-11-00468-t004:** Pros and cons of deformability-based assays.

Pros	Cons
Generally reusable setups	Defects in the matrix can undo separation
More resistant to clogging than DLD	Trapped air bubbles can trap deformable cells
Can theoretically be parallelized	Low throughput due to tiny volumes
Can run even without ratchets	Cannot be run constantly (due to oscillation), unless paired with other approaches e.g., DLD, immuno-magnetic, inertial, or electrokinetic sorting

**Table 5 micromachines-11-00468-t005:** Pros and cons of Dean-flow-based approaches.

Pros	Cons
Generally reusable setups	Comparably big footprint
Can theoretically be parallelized	Prone to clogging
Does not need high resolution lithography Can handle higher cell densities Bigger volumes can be handled	-
Can be run continuously Can be combined with other technologies (e.g., droplet MF)	-

**Table 6 micromachines-11-00468-t006:** Pros and cons of SAW-based approaches.

Pros	Cons
Tunable to different cells by changing frequency	Needs electricity
Work also with bigger, multicellular organisms	Needs piezoelectric and other expensive material
Can be parallelized	Calculations of optimal settings for acoustic forces not yet fully elucidated
Can be run continuously	Small volumes
Can be combined with other techniques	Forces might not fully compete with motile cells

**Table 7 micromachines-11-00468-t007:** Pros and cons of optical tweezer approaches.

Pros	Cons
Contact-free micromanipulation	Needs electricity and lasers & optical setup
Works intracellularly	Relies on differences in refractive indices (e.g., DNA-rich parts like the nucleus)
Can be used to measure forces on the cellular level	Potential photodamage to sample and devices at wrong wavelengths and higher exposures
Can be combined with other techniques	Forces might not fully compete with motile cells
Strong local forces can be achieved An SLM-setup allows cell manipulation in 3D	However, only in “transparent” samples Hard and expensive to parallelize

**Table 8 micromachines-11-00468-t008:** Pros and cons of droplet microfluidics-based approaches.

Pros	Cons
High throughput and parallelization possible	Additional hydrophobic phase needed, plus either detergent (surfactant) or electrowetting
Reactions can be triggered and their process studied	Microfluidic devices are more complex and thus far cannot be brought outside a lab
Entire libraries of drug targets can be screened	-
Droplets can be stored in loops for long term studies	-
Can be combined with other techniques	-

**Table 9 micromachines-11-00468-t009:** Techniques to measure the cell refractive index, sorted by approach.

Approach	Measurement Technique	Refractive Index Resolution	Spatial Resolution (nm)	Minimal Mass Density Change (g/mL)^1^	Ref.
Bulk (Average refractive index of suspended cells)	Interference refractometer	$3\times 10^{-3}$	NA	0.0163	[187]
	Light scattering	$1\times 10^{-2}$	NA	0.0542	[188]
	Light transmission and reflection	$1\times 10^{-2}$	NA	0.0542	[187]
	Optical densitometer	$3\times 10^{-3}$	NA	0.0163	[189]
Single cell level	Fabry-Pérot resonant cavity	$3\times 10^{-3}$	NA	0.0163	[182]
	Grating resonant cavity	$1\times 10^{-3}$	NA	0.0054	[184]
	Immersion refractometer	$1\times 10^{-3}$	NA	0.0054	[190]
	Laser resonant cavity	$4\times 10^{-3}$	NA	0.0217	[180]
	Light scattering	$1\times 10^{-2}$	NA	0.0542	[140]
	Mach-Zehnder Interferometer	$1\times 10^{-3}$	NA	0.0054	[183]
(Sub-)cellular refractive index mapping	Common-path tomographic diffractive microscopy	$1\times 10^{-3}$	NA	0.0054	[191]
	Confocal quantitative phase microscopy	$4\times 10^{-3}$	NA	0.0217	[164]
	Digital holographic microscopy	$3\times 10^{-4}$	NA	0.0016	[163]
		$1\times 10^{-2}$	NA	0.0542	[192]
	Hilbert phase microscopy	$2\times 10^{-3}$	1000	0.0108	[193]
	Microfluidic off-axis holography	$5\times 10^{-3}$	350	0.0217	[194]
	Phase-shifting interferometry	$9\times 10^{-3}$	250	0.0488	[195]
		$3\times 10^{-4}$	NA	0.0016	[196]
	Surface Plasmon nano-optical probe	$4\times 10^{-5}$	80	0.0002	[197,198]
	Tomographic bright-field imaging	$8\times 10^{-3}$	260	0.0434	[199]

^1^ Values were calculated by Liu *et al.* in [185].

**Table 10 micromachines-11-00468-t010:** Pros and cons of cell refractive index-based approaches.

Pros	Cons
Very adaptable to target cells in general	Has to be combined with separation techniques
Generally can be used for high-throughout	Additional setups can be expensive, depending on the technique employed
Depending on the technique, the needed machinery might already be present for the setup	-
Generally damage-free to sample cells	-
Can be combined with other techniques	-

**Table 11 micromachines-11-00468-t011:** Pros and cons of paper microfluidics.

Pros	Cons
Can integrate separation and detection	No continuous phase or separation of cells
Can be used for complete diagnostic chips	Not usable for high throughput
Potentially cheap enough for disposable tests	Parallelization limited
Disposable tests can be combined with analytic cassettes to allow mass field testing with combined lab-based analysis	Slower than liquid–liquid microfluidics
Very mature technology with many applications	Mostly single use devices

**Table 12 micromachines-11-00468-t012:** Pros and cons of PCR/LAMP-based analysis.

Pros	Cons
Reliable	Needs electricity/highly controlled heat source
Can be combined for diagnostic chips	Needs computer hard and software for quantitative tests
Parallelizable and ready for high throughput	Sensitive to impurities and contamination
To some extent, test kits can be pre-treated with reactants to enable field-applicable testing	-
Can be combined with all above separation techniques, including paper microfluidics	-

**Table 13 micromachines-11-00468-t013:** Pros and cons of omics-based analysis.

Pros	Cons
Can give a complete picture of intracellular life	Needs electricity
Can be combined for diagnostic chips	Needs computer hard and software for qualitative and quantitative tests
Generally ready for high throughput	Sensitive to impurities and contamination
Can be used to find out changes (due to stimuli) in complex samples like tissues	Needs highly trained personnel
Can be combined with all above separation techniques, limited with paper microfluidics	Most commercial setups lack the flexibility to adapt the setup (i.e., change eluents) to improve separation of non-standard target molecules

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
