# Peer review of "Lab-on-a-Chip Technologies for the Single Cell Level: Separation, Analysis, and Diagnostics"

_micromachines, 2020, doi:10.3390/mi11050468_

Round 1
Reviewer 1 Report
This manuscript reviews the recent advances towards single cell diagnostic on-a-chip. This is an interesting review but needs some revisions to be qualified for publishing on Micromachines.
Major comments:
1. The significance of the recent research towards the single cell diagnostic on-a-chip should be highlighted in abstract and within the introduction.
- Authors need to make a table and summarize all the published techniques and their applications towards the single cell diagnostic on-a-chip. This will give an easy access to the manuscript’s contents.
- In page 12, authors mentioned about some usage of paper-based microfluidics for single cell diagnostics; however, this type of microfluidic devices have been extensively employed for single cell detection and worth to properly review them.
- For each mentioned method, authors need to discuss the main advantages and shortcomings.
- What happened to the size of the text on section ”2.3.10. Polymerase chain reaction (PCR, nested PCR, qPCR/ RT-PCR)”.
- The applications of each mentioned method should be properly discussed. You need to mention and cite at least 5 different applications for each method.
- Inertia microfluidic devices have been employed for the detection of circulating feta cells in mother’s blood. Please add this application to the right section of the manuscript.
Author Response
Reviewer 1
This manuscript reviews the recent advances towards single cell diagnostic on-a-chip. This is an interesting review but needs some revisions to be qualified for publishing on Micromachines.
Major comments:
- The significance of the recent research towards the single cell diagnostic on-a-chip should be highlighted in abstract and within the introduction.
>The significance has been highlighted in the abstract and the introduction. The introduction was also expanded and adapted in correspondence to the other reviewer’s remarks.
- Authors need to make a table and summarize all the published techniques and their applications towards the single cell diagnostic on-a-chip. This will give an easy access to the manuscript’s contents.
>Table made and added right after the introduction to grant easy access. - In page 12, authors mentioned about some usage of paper-based microfluidics for single cell diagnostics; however, this type of microfluidic devices have been extensively employed for single cell detection and worth to properly review them.
>While I do agree with the reviewer, that paper microfluidics single cell detection test kits are worthy to be properly reviewed, it is also my strong conviction, that adding a full and proper review on them would throw the entire review out of balance. However, I have expanded the section, mentioned that it is one of the most promising techniques towards single-cell diagnostics, and supplied more exemplary applications.
This is also to keep the balance with the changing of the title to “Single cell diagnostic chips and bulky samples”, which resulted from the other reviewer´s comments. - For each mentioned method, authors need to discuss the main advantages and shortcomings.
>I added tables with all the summarized advantages and shortcomings for all methods mentioned, addressing PCR-based and LAMP-based techniques together, since LAMP has evolved from PCS. - What happened to the size of the text on section ”2.3.10. Polymerase chain reaction (PCR, nested PCR, qPCR/ RT-PCR)”.
>Paragraph was in the wrong format. Reformatted according to journal template. - The applications of each mentioned method should be properly discussed. You need to mention and cite at least 5 different applications for each method.
>Additional applications have been mentioned and cited.
For DEP: Smith et al. (2017), Noghabi et al. (2019), and Faraghat et al. (2017).
For deformability-based: Park et al. (2016), Chen et al. (2019), Chen (2018) and Zhou et al. (2016).
For Dean-flows: Zuvin et al. (2016), Zhou et al. (2013), Mach & di Carlo (2010), and Wu et al. (2013).
The other techniques had already at least five different applications mentioned and cited. - Inertia microfluidic devices have been employed for the detection of circulating feta cells in mother’s blood. Please add this application to the right section of the manuscript.
>Two publications reporting the usage of microfluidics for prenatal screening of fetal cells in mother’s blood have been added to the DLD section.

Reviewer 2 Report
This review targets a cutting-edge research topic of single cell chip. My general suggestion to the authors is that they need to be really careful about what is the topic they want to talk about, is it a chip analysing cells in a single-cell level, or a general lab-on-chip system for bulk cell analysis. It seems to me that most technologies discussed here are for general cell isolation and analysis not in a single cell level. Also, the single-cell analysis has been far away from reaching the point for medical diagnostics, so I would not call them a “diagnostic” chip as there are very few examples for a single-cell analysis to be applied in a clinical application yet.
The authors need to restructure the review according to my suggestions above.
If they want to stay with single cell chips, they need to choose the technologies that really talk about single cell isolation (such as droplet microfluidics, and microwells for single cell culturing) and analysis from single-cell level (imaging, single cell flow cytometry, molecular analysis at single-cell level). Also, an important aspect to mention is that WHY single-cell analysis is important. This point has not been mentioned in the manuscript at all. The most important aspect for single cell analysis is for reveal cell heterogeneity. In that case, what information can we get from single cells that can’t be provided by bulk cell analysis? Can the authors give some typical examples? Also, the whole part 3 Implementation should be removed if the authors focus on single-cell level, because so far no POC device is implemented at the single cell level. However, the authors can consider to include the topic of digital detection which is at the single cell/molecule level.
If the authors want to change to bulky cell isolation and analysis (in factor most of the technologies are for bulky cells not for general cells), they can structure the review into sections below:
- Cell isolation methods to separate different types of cells(like DEP, DLD)
- Isolation and analysis in one device
- Molecular analysis after isolation (in a context of lab-on-chip system instead of general molecular analysis discussion such as PCR and proteomics)
Here are some comments on the manuscript itself:
- In the 2nd paragraph in “Introduction”, the author mentioned that many diagnostic devices end in the “valley of death”. I suggest the authors to provide their own opinions why this happens.
- The first paragraph in “Methodologies” need to be removed as it is the template instruction
- Line 101, page 3, the typo 90u should be 90 °.
- Line 172, page 6, the authors mentioned that the ways to tune DLD. Actually the basic ways for that is to change the geometry of DLD structure, which is not mentioned here.
- Line 292, the optical tweezers is an interesting method to manipulate and analyse cells in single cell level. However, the discussion for this part is way too short.
- Line 298, part 2.3.7 is too short without any context in a chip system. It is not appropriate to just mention some basic principle and refer the readers to some other reviews. (In fact this happens several times in this manuscript, just touching the basis and stopped, such as part 2.3.10)
- Part 2.3.10, the format of the first paragraph needs to be changed
- Line 404, the difference of temperature is only mentioned in the caption of Figure 10, not in the main text.
- Line 420, details about the work from Reece et al. should be given if it is an important work in the single cell level.
Author Response
Reviewer 2
This review targets a cutting-edge research topic of single cell chip. My general suggestion to the authors is that they need to be really careful about what is the topic they want to talk about, is it a chip analysing cells in a single-cell level, or a general lab-on-chip system for bulk cell analysis. It seems to me that most technologies discussed here are for general cell isolation and analysis not in a single cell level. Also, the single-cell analysis has been far away from reaching the point for medical diagnostics, so I would not call them a “diagnostic” chip as there are very few examples for a single-cell analysis to be applied in a clinical application yet.
The authors need to restructure the review according to my suggestions above.
If they want to stay with single cell chips, they need to choose the technologies that really talk about single cell isolation (such as droplet microfluidics, and microwells for single cell culturing) and analysis from single-cell level (imaging, single cell flow cytometry, molecular analysis at single-cell level). Also, an important aspect to mention is that WHY single-cell analysis is important. This point has not been mentioned in the manuscript at all. The most important aspect for single cell analysis is for reveal cell heterogeneity. In that case, what information can we get from single cells that can’t be provided by bulk cell analysis? Can the authors give some typical examples? Also, the whole part 3 Implementation should be removed if the authors focus on single-cell level, because so far no POC device is implemented at the single cell level. However, the authors can consider to include the topic of digital detection which is at the single cell/molecule level.
If the authors want to change to bulky cell isolation and analysis (in factor most of the technologies are for bulky cells not for general cells), they can structure the review into sections below:
- Cell isolation methods to separate different types of cells (like DEP, DLD)
- Isolation and analysis in one device
- Molecular analysis after isolation (in a context of lab-on-chip system instead of general molecular analysis discussion such as PCR and proteomics)
>The reviewer raises a valid point here. In order to consolidate the views of all reviewers and of the author, the following steps have been taken:
>The title was adjusted to “Single cell diagnostic chips and bulky samples”.
>The entire text was put into the following sections:
- Cell separation methods
- Combined separation and analysis on chip
- Molecular analysis of single cells
Consequently, the numbering of the subheadings was adapted.
Droplet MF are explained before the techniques relying on optical density.
Droplet MF, Optical density measurements and paper MF are grouped in the section “Combined separation and analysis on chip”, and appropriately adapted.
A table was added to give an overview of all the techniques described in this review.
Here are some comments on the manuscript itself:
- In the 2nd paragraph in “Introduction”, the author mentioned that many diagnostic devices end in the “valley of death”. I suggest the authors to provide their own opinions why this happens.
>Opinions added - The first paragraph in “Methodologies” need to be removed as it is the template instruction
>paragraph removed, and replaced with the actual methodologies of this review. - Line 101, page 3, the typo 90u should be 90 °.
>typo corrected. - Line 172, page 6, the authors mentioned that the ways to tune DLD. Actually the basic ways for that is to change the geometry of DLD structure, which is not mentioned here.
>The basis of how the geometries of the DLD define the cell sorting are now explained as well at his point. Additionally the row shift ε has been introduced 30 lines above this paragraph. - Line 292, the optical tweezers is an interesting method to manipulate and analyse cells in single cell level. However, the discussion for this part is way too short.
>The section on optical tweezers has been vastly expanded, including its discussion. - Line 298, part 2.3.7 is too short without any context in a chip system. It is not appropriate to just mention some basic principle and refer the readers to some other reviews. (In fact this happens several times in this manuscript, just touching the basis and stopped, such as part 2.3.10)
>The section on refractive index measurements was expanded accordingly. It now features a more comprehensive introduction detailing the basics of refractive indices, and several applications of on-chip systems which measure single cells using light scattering (Flynn, Shao, Chachisvilis, Ozkan, & Esener, 2005a), phase contrast microcopy (P. Y. Liu et al., 2014), laser resonant cavity (Liang et al., 2007a), Fabry-Pérot cavity (W. Z. Song et al., 2006b), Mach-Zehnder interferometry (W. Z. Song et al., 2007), or combining an optical trap with a grating resonant cavity (Chin et al., 2007).
In addition, section 2.3.10 was expanded. - Part 2.3.10, the format of the first paragraph needs to be changed
>paragraph reformatted according to journal template. - Line 404, the difference of temperature is only mentioned in the caption of Figure 10, not in the main text.
>Temperatures and details on the amplification loop added - Line 420, details about the work from Reece et al. should be given if it is an important work in the single cell level.
>Details were added detailing the significance of Reece et al.’s work.
>About the general suggestions:
I understand the notion of the reviewer to be careful about the topic and the general idea this review is about. It is true that most technologies are currently working on using bulks of cells to separate them and subsequently analyze several (individual) cells. Also, that on the single-cell level there is quite a lot of heterogeneity, that we (as in the entire scientific community) has not yet fully cartographed and catalogued. Additionally, this heterogeneity (which is now discussed in the introduction) also is the reason why diagnosis can only in very few cases be based on single cells. These cases are exclusive based for invasive pathogenic cells (e.g. parasites like trypanosomes), or endogenous cells that are whose measurable alteration has been caused exclusively by certain negative conditions, and that do not overlap with any other causes.
As for the fact that these technologies so far have not been implemented, yet, this is discussed in more breadth in the introduction (see point 1 above) and in the conclusions. Raising awareness of the underlying causes for this “not having reached maturity, yet” is one of the goals of this review.
I welcome the reviewers suggestion on the re-structuring of the review and have them implemented.
Additionally, to reflect the reviewer’s points, I suggest we change the title of the review to “Single cell diagnostic chips and bulky samples”

Round 2
Reviewer 2 Report
The review is now much improved for publication. But the title is still confusing. How about: "Lab-on-a-chip diagnostics towards single cell separation and analysis" ?
Author Response
Dear Reviewer, dear Editor,
I see your point in the title not fully encompasing the breadth and width of the article. I thus suggest, we change it to:
Lab-on-a-chip technologies for the single cell level: separation, analysis and diagnostics
Thanks and have a great weekend
Axel Hochstetter